# *Lymnaea stagnalis* and *Ophryotrocha diadema* as Model Organisms for Studying Genotoxicological and Physiological Effects of Benzophenone-3

**DOI:** 10.3390/toxics11100827

**Published:** 2023-09-30

**Authors:** Alfredo Santovito, Alessia Pappalardo, Alessandro Nota, Marino Prearo, Dáša Schleicherová

**Affiliations:** 1Department of Life Sciences and Systems Biology, Via Accademia Albertina 13, 10123 Turin, Italy; alessia.pappalard544@edu.unito.it (A.P.); alessandro.nota@conted.ox.ac.uk (A.N.); dasa.schleicherova@unito.it (D.S.); 2IZS PLV (Istituto Zooprofilattico Sperimentale del Piemonte, Liguria e Valle d’Aosta), Via Bologna 148, 10154 Torino, Italy; marino.prearo@izsto.it

**Keywords:** freshwater benthos, marine zooplankton, genotoxicity, micronuclei, egg production, mortality

## Abstract

Benzophenone-3 (BP-3) is a lipophilic organic compound that occurs naturally in flower pigments. Since it adsorbs ultraviolet (UV) radiation in the UVA and UVB regions, it is one of the most common UV filters found in sunscreen and cosmetic products. We explored by in vivo micronuclei (MNi) assay the genotoxic effects of BP-3 on hemocytes from the freshwater gastropod *Lymnaea stagnalis*. We also studied its possible toxic effects on life-history traits: body growth in *L. stagnalis* and egg production of both *L. stagnalis* and the marine polychaete worm *Ophryotrocha diadema.* Adult individuals were exposed to increasing concentrations of BP-3 (0.025, 0.050, 0.100, and 0.200 mg/L) once a week for 4 weeks. In *L. stagnalis,* exposure to BP-3 at concentrations of both 0.2 and 0.1 mg/L produced genotoxic effects on the micronuclei frequencies, but only concentrations of 0.2 mg/L affected the NBUDs frequencies. Similarly, negative effects on body growth were observed at the concentrations of 0.2 and 0.1 mg/L and a significant reduction of egg production at 0.2 mg/L. In *O. diadema*, a negative correlation between egg production and increasing BP-3 concentrations was observed. Our findings suggest the need for more stringent measures to reduce the presence of BP-3 in the environment.

## 1. Introduction

Man-made environmental pollution is one of the most urgent threats to human health and environmental sustainability. Organic ultraviolet (UV) filters are among the many synthetic compounds that go into products for protection against UV radiation, particularly sunscreens, cosmetics, shampoos, and lotions. UV filters are also added to industrial and consumer products (e.g., plastics, food packaging, detergents, insecticides, coatings, printing inks) to increase product longevity [1,2,3].

One of the most widely used UV filters is benzophenone-3 (2-hydroxy-4-methoxybenzophenone; oxybenzone; BP-3), a lipophilic organic compound that occurs naturally in flower pigments. It can adsorb UV radiation in the UVA and the UVB region, with limited phototransformation, properties that make it an ideal ingredient in a variety of sunscreen and other personal care products [4,5,6]. Furthermore, BP-3 is used as a photostabilizer in food packaging materials, plastic surface coatings, and polymers, as well as a component of artificial materials, including textiles and paints [7,8]. Since UV filters are often chemically inert, they escape conventional wastewater treatment and contaminate recycled water systems, natural water bodies, and drinking water resources [9]. Because of inadequate wastewater treatment, BP-3 has been detected at the maximum concentrations of 125 ng/L in freshwater [10,11], whereas concentrations values of 216 ng/L for Italy [12] and 3317 ng/L for Spain [13] were recorded in seawater. The presence of this compound affects water quality and poses risks to human health and aquatic and marine ecosystems [10]. 

From an ecological perspective, the increasing use of BP-3, and of UV filters in general, constitutes a potential risk for the environment. UV filters have been detected in freshwater, coastal ecosystems [14], and aquatic biota. They have been demonstrated toxic to cyanobacteria [15,16], algae [15,17], jellyfish [18], mollusks [19], fish [20], clams, oysters, and intertidal gastropods [21], loggerhead turtles [22], and dolphins [1,2,3,23]. Moreover, since they can enter coral reefs indirectly via municipal wastewater or directly from sunscreen washed off the skin of swimmers, the BP-3 contained in the cream may contribute to coral bleaching and toxicity [24,25]. BP-3 inhibits photosynthesis and induces oxidative damage to tissues in terrestrial plants [26]; moreover, it reduces photosynthetic pigment in microalgae [15].

Owing to its estrogenic and anti-androgenic properties, BP-3 has also been suspected of endocrine-disrupting effects on reproduction in fish [27] and other organisms, such as the crustacean *Daphnia magna* [28], the chlorophyte microalgae *Scenedesmus vacuolatus* [29], and the terrestrial white-tailed eagle *Haliaeetus albicilla* [30]. BP-3 penetrates the skin and enters the bloodstream; it accumulates in adipose tissues, breast milk, and semen, and it is mainly excreted with urine [8]. In humans, it has been linked to reproductive and endocrine disorders, such as reduced couple fecundity and semen quality [31,32], birth outcomes [33], uterine leiomyoma [34], impaired hormonal balance, and low testosterone levels in men [35,36]. 

Finally, genetic and cytogenetic studies have reported that BP-type UV filters induce mutagenic effects in *Salmonella* [37] and sister chromatid exchange and chromosomal aberrations in Chinese hamster ovary cells [38]. In addition, BP-3 was linked to increased micronuclei (MNi) frequency in human lymphocytes [39]. 

In response to growing concern about the adverse effects of BP-3 on human and ecosystem health, the European Commission established a limit of 6% in sunscreen products and 0.5% in other cosmetics [40]. The Council of Europe further set maximum BP-3 levels of 2 mg/kg for foods and 0.5 mg/kg for beverages [41]. According to the European Food Safety Authority, the estimated dietary exposure to BP-3 in the United States and the European Union (EU) is 11 μg and 23 μg per capita per day, respectively [41]. Finally, the EU established a temporary tolerable daily intake (TDI) and an acceptable daily intake (ADI) for BP-3 of 0.1 mg/kg body weight [42]. 

In the present study, we investigated the harmful effects of BP-3, one of the most common UV filters found in sunscreen products. For this purpose, we have chosen to test the effects of this compound on two model organisms present in aquatic environments, one in freshwater (*Lymnaea stagnalis*) and one in the marine environment (*Ophryotrocha diadema*). Our study models are both simultaneous hermaphrodites with rather fast reproductive cycles and short lifetimes. Thus, both models are suitable to study the genotoxic effects of BP-3 as well as its properties as an endocrine disruptor. Here, we evidenced by in vivo micronuclei (MNi) assay the genotoxic effects of BP-3 on hemocytes collected from the freshwater gastropod *L. stagnalis*. Genomic damage was defined as micronuclei, which originate from acentric chromosome fragments or whole chromosomes that fail to segregate correctly during mitotic division and appear as small additional nuclei in the cytoplasm of interphase cells. As such, they demonstrate clastogenic damage due to chromosome/chromatid breaks induced by xenobiotics and aneugenic damage because of agents that interfere with the mitotic apparatus, thus leading to the missegregation of whole chromatids or chromosomes during mitosis. In both instances, the chromatin is not properly distributed to the daughter nuclei and remains in the cytoplasm as a micronucleus. Furthermore, chromosomal instability can be measured by evaluating nuclear buds (NBUDs), which indicate the elimination of amplified DNA and excess chromosomes from aneuploid cells. In addition, we also studied the possible adverse effects of BP-3 on life-history traits such as body growth in *L. stagnalis* and on the allocation of reproductive resources to female function (egg production) in *L. stagnalis* and the marine polychaete worm *O. diadema*. 

We tested four different nominal concentrations of BP-3: 0.20, 0.10, 0.05, and 0.025 mg/L. The concentration at 0.10 mg/L represents the Tolerable Daily Intake (TDI) concentration established by the European Union (0.1 mg/Kg BW) for this compound [41]; 0.20 mg/L is a multiple of the TDI, whereas 0.05 and 0.025 mg/L represent the sub-multiple of this value, tested in order to determine the genotoxicity threshold limit.

## 2. Materials and Methods

### 2.1. Chemicals and Media

The IUPAC name of Oxybenzone (CAS n. 131-57-7) is 2-Hydroxy-4-methoxybenzophenone or BP-3. 

BP-3 and Giemsa stain solution were obtained from Sigma-Aldrich, Milan, Italy. Methanol, acetic acid, and conventional microscope slides were purchased from Carlo Erba Reagenti, Milan, Italy. Potassium chloride (KCl) and Sörensen buffer were obtained from Merck S.p.A., Milan, Italy. Vacutainer blood collection tubes were from Terumo Europe, Rome, Italy.

As for its stability and photodegradation, BP-3 was found to degrade only about 4% after 28 days in water and to remain persistent after 24 h of simulated sunlight irradiation [43].

The chemistry of the tap water used for *L. stagnalis* experiments was as follows: pH: 7.3; dry residue at 180 °C: 313 mg/L; calcium: 71 mg/L; magnesium: 13 mg/L; ammonium: <0.05 mg/L; chlorides: 17 mg/L; sulfates: 35 mg/L; potassium: 2 mg/L; sodium: <10 mg/L; arsenic: <1 mg/L; bicarbonates: 238 mg/L; free residual chlorine: 0.1 mg/L; fluorides: <0.1 mg/L; nitrates: 21 mg/L; nitrites: <0.05 mg/L, and manganese: <1 µg/L. The BP-3 was not found [44]. 

The artificial seawater used for the experiments on *O. diadema* was made in the laboratory starting from deionized water, to which the necessary quantity of salts was added to reach the final salinity of 35‰.

### 2.2. Lymnaea stagnalis

*L. stagnalis* is frequently used as a model organism in ecotoxicological and biological research for studying fundamental mechanisms in neurobiology, evolution, genome editing, omics, and human disease modeling. *L. stagnalis* individuals have a lifespan of about 2 years; they are simultaneous hermaphrodites and sexually mature within 3 months after egg laying. The species is a simultaneous hermaphrodite, meaning that it can both cross- and self-mate. The male reproductive organs are functional before the female ones; males mature, on average, at the age of 30 days, whereas females mature at about 60 days [45] (Figure 1A). During mating, partners regularly alternate sexual roles by reciprocal exchange of sperm (so-called sperm trading) [46]. *L. stagnalis* can be easily maintained in the laboratory; it tolerates a wide range of temperatures (0 to 26–28 °C) and pH (pH 6 to pH 8.5). It is omnivorous and feeds mainly on algae and plants.

### 2.3. Experimental Set-up for Lymnaea stagnalis

*L. stagnalis* individuals involved in the study came from our parasite-free laboratory culture and were reared in the same water and feeding conditions as the experimental groups. In order to avoid any confounding factor involving incomplete sexual maturity, we randomly selected reproductively mature individuals capable of producing eggs. The shell length (measured with a caliber) corresponds to the distance between the apex and the aperture following the central axis.

The experiment was conducted using 10 L plastic containers filled with 8 L of tap water. To reduce evaporation, the containers were closed with a lid, but air could pass via lateral slits on the containers. A single replicate of 20 individuals per concentration was used. For *L. stagnalis,* we decided not to divide the animals into replicates. This procedure is not generally required for genotoxicology studies; we also wanted to align in vivo studies with in vitro ones conducted on human lymphocytes, for which no replications are required.

To test the effects of BP-3 exposure, defined as genomic damage, reduced body growth, and egg production, five groups were randomly formed of young adults with a shell length (i.e., distance from the apex to the aperture of the shell, following the central axis) ≥20 mm; this was done to exclude incomplete sexual maturity or inability to lay eggs (Table 1):

In order to obtain the final concentrations of 0.2, 0.1, 0.05, and 0.025 mg/L, we dissolved 1.6, 0.8, 0.4, and 0.2 g of BP-3 in 8 L of water, respectively. The negative control was represented by water without BP-3. 

Individuals were reared at room temperature (range, 18–22 °C) and fed the same amount of food (100 g vegetables/week, mainly salad). All subjects involved in the study were randomly selected from our parasite-free laboratory culture, where rearing conditions were uniform (i.e., tap water, temperature range between 18 and 22 °C, and salad as a primary source of food). This was done to exclude any effect of biological variability due to individual-specific responses to the changing water conditions. During the 4 week study period, water, food, and BP-3 concentrations were renewed twice a week for each group. All the experimental groups were kept under the same light-dark regime. The number of eggs laid was recorded each week; changes in shell growth (length in mm) were measured at the beginning and at the end of the experiment. 

### 2.4. Micronuclei Assay on Hemocytes from Lymnaea stagnalis 

The sample was 20 individuals per group exposed for 4 weeks to increasing concentrations (0.2, 0.1, 0.05, and 0.025 mg/L) of BP-3. At 4 weeks of xenobiotic exposure, hemolymph was collected by stimulating its release by prodding the animal’s foot with a micropipette (Figure 1B). Five hundred microliters of hemolymph per subject were collected and distributed onto clear microscope slides. The cells were then fixed by adding several drops of methanol/acetic acid solution in a 3:1 ratio. The slides were dried, and the cells were stained for 10 min by a conventional staining method using 5% Giemsa (pH 6.8) prepared in Sörensen buffer. The slides were washed with distilled water and dried; the cells were observed under a Leica Dialux 20 light microscope (magnification 1000×). We analyzed 1000 hemocytes with intact nuclear and cellular membranes per subject per concentration, and the number of MNi was scored. MNi were identified according to the following criteria: diameter < 1/3 of the main nucleus; coloring and refractive index similar to the main nucleus; absence of direct contact between MNi and the main nucleus.

### 2.5. Ophryotrocha diadema

*O. diadema* (Dorvilleidae) (Figure 2), a marine polychaete worm measuring 2–3 mm in length, is a simultaneous hermaphrodite with external fertilization. Data on mating [47] and life cycle [48] have been obtained by laboratory observation. The simultaneously hermaphroditic phase, preceded by the protandrous phase, starts when individuals reach 14–15 segments in length and lasts for about 30–40 days. Short courtship is followed by mating achieved by pseudocopulation and external fertilization. During external fertilization, the partners are in close physical contact with each other before releasing their gametes. During mating, the partners regularly alternate the female and the male sexual roles every 2 to 3 days by reciprocal exchange of transparent cocoons. This reproductive strategy is termed egg trading. Each cocoon contains about 30 eggs and can be easily viewed under a stereomicroscope. The sperm in *O. diadema* are immotile [49]. Nine days after egg laying, the offspring hatch as small four-segment larvae with viable sperm. The maximum body length of *O. diadema* is 20 to 21 segments, and its lifespan is 90 days. The species exhibits two different phenotypes: white (yy) lay white eggs, and yellow (Yy, YY) lay yellow eggs. *O. diadema* populations belong to marine zoobenthic species of fouling fauna inhabiting Californian harbors. Despite their low population density, the adults produce a network of mucous trails and can be easily followed by conspecifics, which likely makes for a clustered spatial distribution. 

We utilized individuals of an *O. diadema* strain derived from individuals collected by Prof. B. Åkesson (University of Stockholm) in 1976 and 1980 in the Californian harbors (Long Beach). In order to increase genetic variability and to refresh our laboratory populations, new individuals were added to the strain in 1995 and again in 2001, 2006, and 2021. These animals came from other laboratory cultures and were kindly sent by Prof. B. Åkesson, Prof. R. Simonini (University of Modena), and Prof. M. C. Lorenzi (University of Paris).

### 2.6. Experimental Set up for Ophryotrocha diadema

Our laboratory’s *O. diadema* strains have been cultured for years under the same conditions: T = 20 °C, artificial seawater salinity = 35 ‰, and feeding with parboiled spinach. The experiment was carried out in 100-mL glass bowls filled with artificial seawater. The above-mentioned laboratory conditions were maintained during the experiment. 

In order to reduce evaporation, all bowls were placed in closed boxes in a thermostatic cabinet maintained at 20 °C. Dissolved oxygen and temperature were kept constant during the study period. Since the experiment was set up to test the effects of BP-3 exposure on female reproductive output in *O. diadema*, five groups were randomly formed of adult virgin hermaphrodites of the same age, no siblings, (offspring (F1) of 24 *O. diadema* parent pairs (PP)) (Table 2). Eleven replicates per group were performed.

We prepared a stock solution of BP-3 at a concentration of 2 mg/L. The stock solution was subsequently diluted 10, 20, 40, and 80 times to reach the final concentrations of 0.2, 0.1, 0.05, and 0.025 mg/L used during the experiment. The negative control was represented by water without BP-3. During the 4 week study period, mortality and female reproductive parameters (number of cocoons and eggs/cocoon) were recorded twice a week.

### 2.7. Statistical Analysis 

Statistical analysis was performed using IBM-SPSS 28.0 for Windows (IBM-SPSS, Armonk, NY, USA). All *p* values are two-tailed; *p* values ≤ 5% are considered statistically significant for all tests. The Shapiro–Wilk test was performed to assess data distribution normality. The homogeneity of variance was tested using the Levene test. We compared the mean number of MNi, NBUDs, bi-nucleated hemocytes, number of eggs laid, and differences in body growth between the treated groups and the controls by means of non-parametric Mann–Whitney tests or ANOVA according to data distribution. The chi-square test was used to evaluate differences in the number of eggs laid by *L. stagnalis* exposed to four different BP-3 concentrations. 

For *Ophryotrocha*, since the distribution of variances was not homogeneous (Levene’s test; *p* < 0.05) and the number of replicates was quite small (11 per group), non-parametric statistics were preferred for our data analysis. Therefore, the mean number of eggs laid between the groups exposed to different BP-3 concentrations and the controls was compared using the non-parametric Kruskal–Wallis H test. The number of eggs laid was the dependent variable, and the treatment (controls and groups exposed to four different concentrations of BP-3) was the fixed factor. 

## 3. Results

### 3.1. Lymnaea stagnalis

In *L. stagnalis*, the Shapiro–Wilk normality test performed on MNi assay results and life-history traits showed that MNi, NBUDs, binucleated cells, and body growth resulted not normally distributed (*p* < 0.001 for all biomarkers); therefore, the Mann–Whitney test was used to compare genomic damage and body growth in the groups exposed to BP-3. Since the distribution of eggs laid per week was normal (*p* = 0.374), ANOVA was performed to test differences between the groups exposed to BP-3. 

Comparison between the BP-3 concentrations and the biomarkers showed an increased frequency of MNi in *L. stagnalis* hemocytes exposed to 0.2 and 0.1 mg/L BP-3; the latter is the acceptable daily intake in humans established by the EU. The number of NBUDs was increased only in the group exposed to 0.2 mg/L BP-3, whereas no significant difference in the number of binucleated cells was recorded between the groups exposed to BP-3 (Table 3).

Figure 3 shows examples of MNi and NBUDs in *L. stagnalis* hemocytes.

Similar to the results of the MNi assay, body growth was significantly reduced in the groups exposed to 0.2 and 0.1 mg/L BP-3 (Table 4); significant differences in the number of eggs were noted for the group exposed to 0.2 mg/L BP-3 (Table 5).

### 3.2. Ophryotrocha diadema

In the control group, the mean number of eggs laid was 444.18 ± 79.39. A decrease in egg production was noted for the groups exposed to increasing BP-3 concentrations. The mean number of eggs was 350.09 ± 79.65 eggs in the group exposed to the lowest BP-3 concentration (0.025 mg/L), 297.73 ± 30.43 eggs in the group exposed to 0.050 mg/L, 222.64 ± 41.68 eggs in the group exposed to 0.1 mg/L, and 47.27 ± 18.20 eggs in the group exposed to 0.2 mg/L. The Kruskal–Wallis H test showed a significant effect of BP-3 on egg production in all five groups (H = 45.4489, df 4; *p* < 0.001; Figure 4) and between the groups (Table 6).

After Week 2 of treatment, all eggs exposed to the two highest BP-3 concentrations (0.2 and 0.1 mg/L) were found to have degenerated: eggs were laid but did not develop and consequently got dissoluted. Moreover, toxic effects of the highest BP-3 concentration (0.2 mg/L) were observed as a higher mortality rate of adult individuals (47.27%, 52 individuals died/110). At the end of the experiment, we estimated the number of individuals present and alive in each replicate of the five treatments. At the highest concentration (0.2 mg/L), the number of individuals decreased from 110 to 52, while in the other treatments, including controls, the mortality rate was constant. 

## 4. Discussion

Understanding the impact of environmental pollutants in relation to environmental sustainability is sometimes difficult to grasp due to the many interrelated variables, among which are exposure rate and duration, metabolism of chemicals in exposed individuals, and interactions with other environmental stress factors. Wildlife often presents signs of the effects of xenobiotic compounds, most of which have known endocrine-disrupting properties. Through experimental studies on animal models, we may better understand the mechanisms of action of many chemicals at the genomic and physiological level and evaluate the effects of exposure dose and duration. 

BP-3, the most frequently used UV radiation filter in many skincare products, has been detected in the biological fluids of exposed individuals [50,51]. Although readily metabolized in animal tissues, BP-3 is routinely detected in water and soil at high enough concentrations to harm aquatic species and ecosystems. BP-3 has been demonstrated to be toxic to cyanobacteria, green algae, and corals and disruptive to reproduction in fish [3]. 

Here, we studied the genotoxic and toxic effects of four concentrations of BP-3 in two animal study models, i.e., *L. stagnalis*, a freshwater gastropod, and *O. diadema*, a marine polychaete. Our findings show that exposure to BP-3 can induce genomic and reproductive damage at a concentration of 0.1 mg/L, the acceptable daily intake for humans [38]. Documented evidence for cytogenetic damage, also observed at a concentration of 0.10 mg/L, urges setting safer limits of BP-3 for human health. 

The effects of genotoxicity we observed after exposure to 0.1 mg/L BP-3 also call for attention regarding ecosystem health. A previous study reported that a concentration of ≤0.5 mg/L (≤0.5 mg/L) BP-3 was found to affect the sex ratio and the development of gonads in the zebrafish *Danio rerio* [52], whereas a study on the freshwater fish *Betta splendens* found a decrease in mature spermatozoa in testicular tissue after exposure to 0.1 mg/L (0.1 mg/L) [53]. Finally, exposure to 0.1 mg/L BP-3 led to tissue bleaching in the adult coral *Seriatopora caliendrum* [54]. 

The genotoxic properties of BP-3 could be due to its demonstrated ability to induce oxidative stress, probably by decreasing antioxidant enzyme activity and increasing lipid peroxidation [55]. Hanson et al. [56] showed that BP-3 and other UV filters can penetrate the stratum corneum of the skin and generate high ROS content in the cytoplasm of epidermal nucleated keratinocytes. Moreover, the expression of genes encoding heat shock proteins, phase I cytochrome P450 genes, and genes related to oxidative defenses was noted to be affected in fish after exposure to BP-3 [57,58,59,60,61]. Furthermore, the demonstrated accessibility of BP-3 to labile hydrogens within DNA [62] may explain the increased levels of genomic damage induced by BP-3 at concentrations of 0.2 and 0.1 mg/L. Genomic damage is linked to an increased risk of cancer development [63]. Phiboonchaiyanan et al. [64] showed that exposure to BP-3 can increase metastatic potential in lung cancer cells.

We observed reduced body growth of *L. stagnalis* after exposure to concentrations of 0.2 and 0.1 mg/L (Table 4), while differences in the mean number of eggs laid/week were found only after exposure to 0.2 mg/L (Table 5). These observations are shared by Im et al. [65], who observed a reduction in body length and egg production of *Daphnia magna* after exposure to a concentration of 0.8 mg/L. The study suggested that a reduction in the number of offspring may be attributed to a delay in time to first brood and inhibition of embryonic development. 

Our data show toxic effects on the egg production of *O. diadema* after exposure to all four BP-3 concentrations. Egg production was low even at the lowest concentration (0.025 mg/L). In addition, after week 2 of treatment, all eggs exposed to the two highest BP-3 concentrations (0.2 and 0.1 mg/L) were found to have degenerated, indicating toxicity following chronic exposure to BP-3 and affecting embryogenesis. Moreover, a toxic effect of the highest BP-3 concentration was observed in the higher mortality rate (47.27%, 52/110) of individuals exposed to 0.2 mg/L for 4 weeks, whereas the mortality rate was far lower (5.46%) in the controls and in the three other treated groups. 

Similar toxic effects of BP-3 on reproduction have been observed in different target species. For example, exposure to ≤0.5 mg/L of BP-3 skewed the sex ratio towards more females and fewer males in *Danio rerio* [52]. Exposure to BP-3 altered the expression of some sex hormone-related genes, decreased egg production after exposure to 26 mg/L, and increased plasma testosterone levels in males after exposure to 90 mg/L in Japanese medaka (*Orzias latipes*) [11].

It is difficult to predict which impacts could generate these concentrations of BP-3 on the natural populations of the two organisms or on other similar species, also considering the possible synergistic effect with different xenobiotics present in natural environments, including other endocrine disruptors and pesticides. However, here we analyzed the effects of BP-3 at the genomic level and on two fitness parameters, body length, and egg production, that could have consequences at a population level. Indeed, if a xenobiotic induces genomic instability and a decrease in the body growth rate and in number of eggs laid, then the number of adults able to reproduce will decrease as a consequence. Numerous studies demonstrated that growth inhibition in aquatic invertebrates may be associated with a reduction in reproductive outputs. For example, Bessa da Silva et al. [66] showed the negative effect of dietary exposure to herbicide on the reproductive output of *Daphnia magna*. Similarly, Henry et al. [67] demonstrated the negative effects of fluoxetine on the reproduction of freshwater snail *Physa acuta*, whereas Cole et al. [68] evidenced the negative impact of microplastic on function and fecundity in the marine copepod *Calanus helgolandicus*, with consequences on the population dynamics of this species. 

## 5. Conclusions

Here, we described the genotoxic potential and the adverse effects BP-3 can have on reproduction and body growth in two animal study models after exposure to sublethal doses. Our data add to previous findings for the risks to the *L. stagnalis* and *O. diadema* populations as well as for the entire biome that BP-3 exposure can cause. BP-3 was also found to be an endocrine disruptor and, as such, poses a threat to human and wildlife health, biodiversity, and environmental sustainability in general. In this scenario, we suggest that more stringent measures should be adopted to reduce the presence of BP-3 in the environment and to mitigate the effects on health and the environment associated with the discharge of BP-3 and other endocrine disruptors from wastewater. In addition, the search for new compounds to replace BP-3 and other environmental xenobiotics will need to take into account their environmental sustainability, the possible repercussions on human health, and the short and long-term consequences such compounds can have on entire ecosystems.

## Figures and Tables

**Figure 1 toxics-11-00827-f001:**
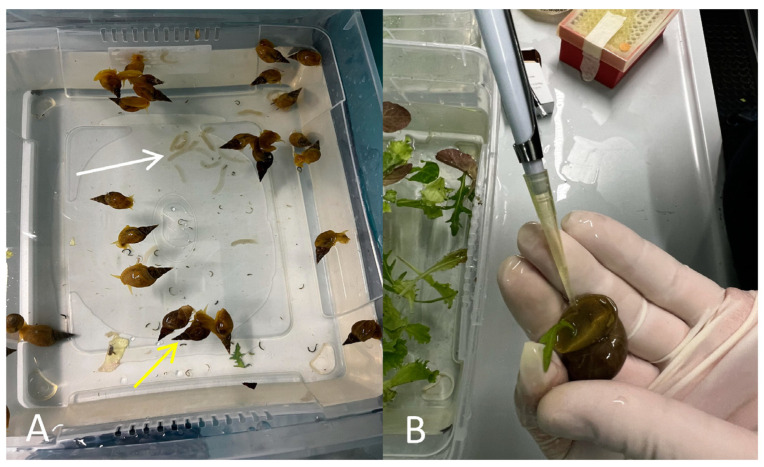
(**A**) Young adult *Lymnaea stagnalis* raised in our laboratory (yellow arrow) and eggs (white arrow). (**B**) Extracting hemolymph by prodding with a pipette.

**Figure 2 toxics-11-00827-f002:**
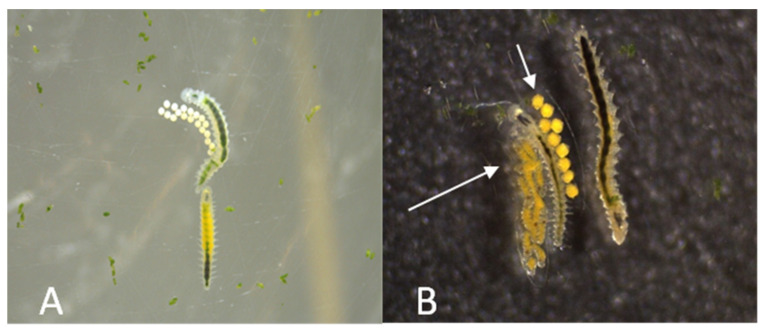
*Ophryotrocha diadema* (family Dorvilleidae) (56× magnification, Leica stero microscope, Leitz, Germany). (**A**): 2 adult individuals, one belonging to the white strain (yy) and one to the yellow strain (Yy), who are doing parental care of the newly laid cocoon; (**B**) 2 yellow adults (Yy), 1 cocoon with yellow eggs (shorter arrow) and one cocoon with developing larvae (longer arrow).

**Figure 3 toxics-11-00827-f003:**
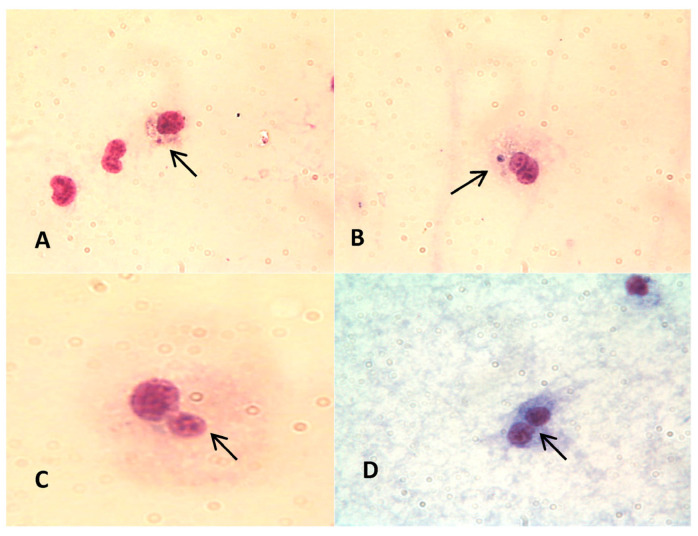
Micronuclei (**A**,**B**), nuclear buds (**C**), and binucleated cells (**D**) in hemocytes from *L. stagnalis* (1000× magnification, Leitz Dialux 20 microscope, Leitz, Germany). The arrows indicate the presence of MNi (**A**,**B**), Nuclear Buds (**C**) and a binucleate cell (**D**).

**Figure 4 toxics-11-00827-f004:**
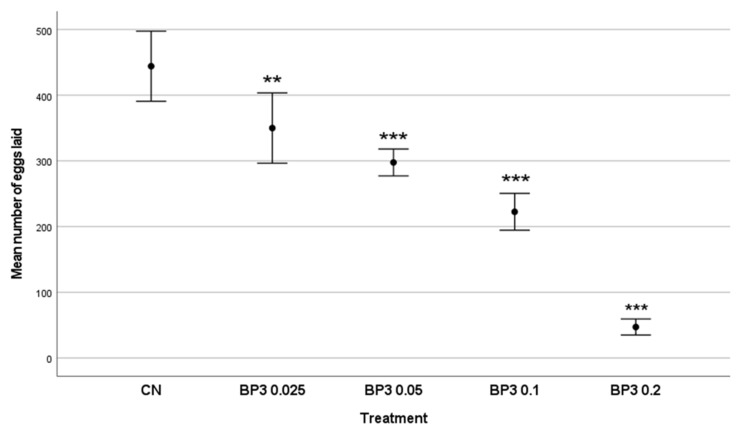
Effect of BP-3 exposure on mean number of eggs laid by *O. diadema* in the control and the four groups. The reproductive output only of living individuals was considered. CN denotes the control negative group; ** *p* = 0.017; *** *p* < 0.001, with respect to the CN group.

**Table 1 toxics-11-00827-t001:** Experimental set-up for *L. stagnalis*.

Group—20 Individuals Per Container	Treatment [BP-3]
Controls	0.000 mg/L freshwater
Group 1	0.200 mg/L
Group 2	0.100 mg/L
Group 3	0.050 mg/L
Group 4	0.025 mg/L

**Table 2 toxics-11-00827-t002:** Experimental set up for *O. diadema*.

Group—10 Individuals Per Bowl	Treatment [BP-3]
Controls	0.000 mg/L pure artificial seawater
Group 1	0.200 mg/L
Group 2	0.100 mg/L
Group 3	0.050 mg/L
Group 4	0.025 mg/L

**Table 3 toxics-11-00827-t003:** Induction of micronuclei in hemocytes (total cells scored: *n* = 20,000) from *L. stagnalis* exposed to four different concentrations of BP-3.

BP-3 (mg/L)	No. of Observed MNi	‰ MNi/Total Cells Scored ± SD	No. of Observed NBUDs	‰ NBUDs/Total Cells Scored ±SD	BNCs	‰ BNCs/Total Cell Scored ±SD
NC	6	0.300 ± 0.470	20	1.000 ± 1.026	8	0.400 ± 0.60
0.025	8	0.400 ± 0.598	21	1.050 ± 0.945	10	0.500 ± 0.83
0.050	9	0.450 ± 0.686	23	1.150 ± 1.040	13	0.650 ± 0.99
0.100	18	0.900 ± 0.852 ***	28	1.400 ± 1.536	12	0.600 ± 1.19
0.200	25	1.250 ± 0.910 *	44	2.200 ± 1.963 **	10	0.500 ± 0.61

MNi denotes micronuclei; NBUDs nucleoplasmic buds; BNCs bi-nucleated cells; NC negative control. * significantly higher compared to NC (Mann–Whitney test; *p* < 0.001); ** significantly higher compared to NC (Mann–Whitney test; *p* = 0.039); *** significantly higher compared to NC (Mann–Whitney test; *p* = 0.013).

**Table 4 toxics-11-00827-t004:** Body length (in mm) of *L. stagnalis* before and after 4 weeks of treatment with different concentrations of BP-3.

	NC Mean ± SD	Group 4 0.025 mg/L Mean ± SD	Group 3 0.05 mg/L Mean ± SD	Group 2 0.10 mg/L Mean ± SD	Group 1 0.20 mg/L Mean ± SD
Before treatment	32.73 ± 1.55	32.45 ± 1.92	32.40 ± 1.47	32.68 ± 1.78	32.60 ± 1.78
After 4 weeks of treatment	49.23 ± 2.83	48.60 ± 268	48.34 ± 2.44	46.39 ± 2.57 **	44.23 ± 3.16 *
Differences	16.51 ± 2.90	16.15 ± 1.98	15.94 ± 2.19	13.71 ± 2.35 °°	11.61 ± 2.17 °

SD denotes standard deviation; NC negative control; BP-3 benzophenone-3; Whitney test; *p* < 0.001); * significantly lower compared to NC, 0.025 and 0.05 BP-3 concentrations (ANOVA test; *p* < 0.001); ** significantly lower compared to NC (ANOVA test; *p* = 0.015); ° significantly lower compared to NC, 0.025 and 0.05 BP-3 concentrations (Mann–Whitney-test; *p* < 0.001); °° significantly lower compared to NC (Mann–Whitney-test; *p* = 0.005).

**Table 5 toxics-11-00827-t005:** Weekly and total number of eggs laid by *L. stagnalis* in the controls (*n* = 20) and groups (*n* = 20 per group) exposed to BP-3. The mortality rate was 0 for each treatment.

	NC	Group 4 (0.025 mg/L)	Group 3 (0.05 mg/L)	Group 2 (0.10 mg/L)	Group 1 (0.20 mg/L)
Week 1	2548	1010	1059	2476	2042
Week 2	1529	1764	1652	964	908
Week 3	2756	1988	1048	1422	764
Week 4	3128	2756	1406	2188	1854
Total	9961	8550	6582	5633	4536
Mean ± SD/week	2490 ± 684	2138 ± 430	1645 ± 607	1408 ± 556	1134 ± 490 *

SD denotes standard deviation; NC negative control; * significantly lower compared to NC (ANOVA; *p* = 0.038).

**Table 6 toxics-11-00827-t006:** Effect of BP-3 treatment on egg production in *O. diadema*.

Between-Group Comparison	Kruskal-Wallis H Test		
	H	df	*p*-Value
Controls vs. Group 4 (0.025 mg/L)	5.745	1	=0.017
Controls vs. Group 3 (0.050 mg/L)	13.765	1	<0.001
Controls vs. Group 2 (0.100 mg/L)	15.783	1	<0.001
Controls vs. Group 1 (0.200 mg/L)	15.783	1	<0.001
Group 1 (0.200 mg/L vs. Group 4 (0.025 mg/L)	15.783	1	<0.001
Group 1 (0.200 mg/L) vs. Group 3 (0.050 mg/L)	15.783	1	<0.001
Group 1 (0.200 mg/L) vs. Group 2 (0.100 mg/L)	15.783	1	<0.001
Group 2 (0.100 mg/L vs. Group 4 (0.025 mg/L)	12.574	1	<0.001
Group 2 (0.100 mg/L vs. Group 3 (0.050 mg/L)	11.885	1	<0.001

## Data Availability

The analytical data will be made available to interested parties upon request.

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
