# Peer review of "Lymnaea stagnalis and Ophryotrocha diadema as Model Organisms for Studying Genotoxicological and Physiological Effects of Benzophenone-3"

_toxics, 2023, doi:10.3390/toxics11100827_

Round 1
Reviewer 1 Report
In this manuscript the authors report on the effects on a UV filter (Benzophenone-3) on genotoxicity, growth and reproduction in the pond snail (L. stagnalis) and reproduction in a marine polychaete worm (O. diadema). The rationale for the doses used in to use the human TDI, and some doses on either side, to look at genotoxic thresholds and reproductive effects.
1. The abstract is well written but is also lacking in specific detail, such as the nominal concentration/exposure that affected the measured endpoints, and the magnitude of the observed effects.
Introduction:
2. Line 40: assume you mean chemically inert
3. Line 44: please include more information on actual measured concentrations in freshwater and marine environments. This is needed to put the exposures into further context.
4. Line 94: you mention 0.0125 ug/ml as an exposure level, but none of the data support this.
Materials and Methods:
There are some issues here to address:
5. L. stagnalis exposure does not have any replicates. Each condition is simply one tank with 20 snails.
6. A serious omission (for both exposures) is that there is no water chemistry. Therefore you cannot prove that you have BP-3 in the water, or that it is at the concentration that you claim. To address this now, I suggest that you (i) include some information in the manuscript about the stability of BP-3 in freshwater and artificial sea water, and/or (ii) do a stability study yourself over 4 weeks with and without animals (even better). NOTE: you must always refer to concentrations as ‘nominal values’ in the absence of analytical chemistry. In addition, if you made a mistake (contamination) you will not be aware of this without analytical chemistry of the exposure water.
7. Please include information about how stock solutions were prepared, and information on the tap water chemistry (including pH).
8. Exposures to Ly. stagnalis were carried out in plastic tubs. As plastic can leach various chemicals, and may also contain BP-3, you need to provide more information about the containers you used. Again, plastic should have been avoided, as chemicals can stick to it, and is not as inert as glass. This should also be discussed in the paper.
9. More information needed on feeding. 100g of vegetables (per day?) seems to be a lot of food, and this could impact water quality.
10. How much hemolymph was extracted from snails and aliquoted to slides for MNi assays? It looks like you assessed this for each of the 20 snails in each treatment, and scored 1000 cells in each case. Therefore you will have 20 data points for MNi for each treatment, and 20,000 cells in total counted per treatment. This also has implications for reporting (Table 3) as explained below.
11. Line 138 – please state hemolymph was collected from each snail.
12. For both the snail and the polychaete there is no acclimation or baseline period, so it is unclear how variable the different groups are in their reproductive output. It is possible that many differences observed are simply a consequence of random biological variability. If this is unlikely, then please explain how adjustments were made to control for biological variability at the outset of the exposure.
Results:
13. Make sure all mention of concentrations refers to nominal (not actual) concentrations.
14. Table 3. This table is confusing when you consider the methodology and approach used. For example, out of the 20 snails you took hemolymph from (and counted 1000 cells) you found a total aggregate of 6 MNi (or 6 cells out of 20000 that you evaluated). If so, then the number of MNi/cell would be 0.0003 MNi/cell (not 0.3MNi/cell as shown). Please check this table carefully.
15. Mortality data is not presented. This is important, because you must adjust reproductive output to the number of surviving animals in each treatment and/or replicate.
16. Table 4. Please remove this table, as we do not need to see data for each individual snail. Instead please show the Mean body length and SD for each condition at week 0 (start) and Week 4 (end). Carry out statistics to determine if body length was different at the start of the exposure between any of the conditions.
17. Table 5 – please state in the legend if there was any mortality. If there is, you must adjust for this in your analysis (eggs/snail).
18. Like the L. stagnalis experiment, the polychaete exposure is also lacking acclimation and baseline readings, so there is the question of influence of biological variability. The photo of polychaete reproduction (Figure 2) suggests it is quite variable (based on numbers of eggs per cocoon shown).
19. There is a mention of mortality occurring in some treatments in the polychaete exposure (line 303 states 47% mortality). Please include mortality data.
20. Please explain if the data presented in Figure 4 has adjusted for mortality. If not please do so. I would advise that you replot Figure 4 so that it shows mean cumulative fecundity and SD, as an aggregate (i.e. shows the total number of eggs produced over time for each condition/treatment).
Author Response
REVIEWER 1
The authors would like to thank the anonymous reviewer for the careful revision, which helped us improve the manuscript.
In the text, changes have been highlighted in red.
1. Reviewer 1: “The abstract is well written but is also lacking in specific detail, such as the nominal concentration/exposure that affected the measured endpoints, and the magnitude of the observed effects”.
Authors: we modified the abstract adding the required information.
Introduction:
2. Reviewer 1: “Line 40: assume you mean chemically inert”
Authors: we added “chemically” before inert.
3. Reviewer 1: “Line 44: please include more information on actual measured concentrations in freshwater and marine environments. This is needed to put the exposures into further context”.
Authors: in the introduction session, we included the concentration value observed for BP-3. In particular, we stated: “Because of inadequate wastewater treatment, BP-3 has been detected at the maximum concentrations of 125 ng/L in freshwater [10, 11], whereas for seawater concentrations values of 216 ng/L for Italy [12] and 3317 ng/L for Spain [13] were recorded. The presence of this compound, although at low concentrations, impacts on water quality and poses risks to human health and aquatic and marine ecosystems [10].
4. Reviewer 1: “Line 94: you mention 0.0125 ug/ml as an exposure level, but none of the data support this”.
Authors: we apologize, it was a mistake. In the revised version we deleted the concentration of 0.0125 ug/ml.
Materials and Methods:
5. Reviewer 1: “L. stagnalis exposure does not have any replicates. Each condition is simply one tank with 20 snails”.
Authors: For Lymnaea, we did not make replicates because they are not generally required for genotoxicology studies. This serves to align in vivo studies with in vitro ones, conducted, for example, on human lymphocytes for which no replications are required. Instead, for the Ophryotrocha, as there is no genotoxic data, we decided to do the replications, as required.
6. Reviewer 1: A serious omission (for both exposures) is that there is no water chemistry.
Authors: For the breeding of Lymnaea stagnalis we used municipal water and based on the data reported by the competent authorities, the presence of BP-3 was not reported. In the corrected version of the manuscript we included the following statement: “The chemistry of the tap water used for L. stagnalis experiments was as follows: pH: 7.3; Dry residue at 180°C: 313 mg/L; Calcium: 71 mg/L; Magnesium: 13 mg/L; Ammonium: <0.05 mg/L; Chlorides: 17 mg/L; Sulphates: 35 mg/L; Potassium: 2 mg/L; Sodium: <10 mg/L; Arsenic: <1 mg/L; Bicarbonates: 238 mg/L; Free residual chlorine: 0.1 mg/L; Fluorides: <0.1 mg/L; Nitrates: 21 mg/L; Nitrites: <0.05 mg/L and Manganese: <1 µg/L. The BP-3 was not found [41].
The artificial sea water used for experiments on O. diadema was made in the laboratory starting from deionized water, to which the necessary quantity of salts was added to reach the final salinity of 35‰”. In the corrected version of the manuscript, we have included these considerations, as well as the stability values of BP-3 in freshwater and seawater. We included this information in Material and Methods session, sub-paragraph 2.1.
7. Reviewer 1: “Therefore you cannot prove that you have BP-3 in the water, or that it is at the concentration that you claim. To address this now, I suggest that you (i) include some information in the manuscript about the stability of BP-3 in freshwater and artificial sea water, and/or (ii) do a stability study yourself over 4 weeks with and without animals (even better)”.
Authors: As for its stability and photodegradation, BP-3 was found to degrade only about 4% after 28 days in water and to remain persistent after 24 h of simulated sunlight irradiation (Imamović et al., 2022).
We included this statement in Material and Method session, sub-session 2.1 Chemicals and Media.
Imamović, B.; Trebše, P.; Omeragić, E.; Bečić, E.; Pečet, A.; Dedić, M. Stability and Removal of Benzophenone-Type UV Filters from Water Matrices by Advanced Oxidation Processes. Molecules 2022, 27(6), 1874. doi: 10.3390/molecules27061874
8. Reviewer 1: “NOTE: you must always refer to concentrations as ‘nominal values’ in the absence of analytical chemistry. In addition, if you made a mistake (contamination) you will not be aware of this without analytical chemistry of the exposure water”.
Authors: At the end of the introduction session, reporting the BP-3 concentration we tested, we put the term “nominal” before the word concentration. Unfortunately, we did not do a chemical analysis of the water during the experiment.However, the breeding conditions were identical for all groups, in terms of water quality and population density.
9. Reviewer 1:“Please include information about how stock solutions were prepared, and information on the tap water chemistry (including pH)”.
Authors: In order to obtain the final concentrations of 0.2, 0.1, 0.05 and 0.025 mg/L we dissolved 1.6, 0.8, 0.4 and 0.2 grams of B-3 in 8L of water, respectively.
For O. diadema, we prepared a BP-3 stock solution at concentration of 2 mg/L. The stock solution was subsequently diluted 10, 20, 40 and 80 times to reach the final concentrations of 0.2, 0.1, 0.05 and 0.025 mg/L used during the experiment.
In the materials and methods, we specified the above.
10. Reviewer 1: “Exposures to L. stagnalis were carried out in plastic tubs. As plastic can leach various chemicals, and may also contain BP-3, you need to provide more information about the containers you used. Again, plastic should have been avoided, as chemicals can stick to it, and is not as inert as glass. This should also be discussed in the paper”.
Authors: It is true, that for the experiments on L. stagnalis we used plastic containers that can potentially release various chemicals. However, we would like to underline that at room temperature (around 20°C) the release of chemical substances is low and that the same type of container was also used for the control group. Therefore, all groups analyzed in the study had the same “background” of potentially toxic substances released from the container and the same chemical composition of the water.
However, we have included this information in the paper in Material and Method session, sub-session 2.3.
11. Reviewer 1: “More information needed on feeding. 100g of vegetables (per day?) seems to be a lot of food, and this could impact water quality”.
Authors: No, they are 100 g/week. In the revised version, we better specified it.
After 1 week, no food residues were observed, demonstrating that the food was completely consumed by the animals.
12. Reviewer 1: “How much hemolymph was extracted from snails and aliquoted to slides for MNi assays? It looks like you assessed this for each of the 20 snails in each treatment, and scored 1000 cells in each case. Therefore you will have 20 data points for MNi for each treatment, and 20,000 cells in total counted per treatment. This also has implications for reporting (Table 3) as explained below”.
Authors: the quantity of haemolymph extracted for all subject was 500 uL. In this quantity there are more than 1000 hemocytes; however we observed the first 1000 cells, after which we stopped. We have included this data in the materials and methods.
13. Reviewer 1: “Line 138 – please state hemolymph was collected from each snail”.
Authors: yes, 500 uL of hemolymph was collected from each snail.
14. Reviewer 1: “For both the snail and the polychaete there is no acclimation or baseline period, so it is unclear how variable the different groups are in their reproductive output. It is possible that many differences observed are simply a consequence of random biological variability. If this is unlikely, then please explain how adjustments were made to control for biological variability at the outset of the exposure.”
Authors: L. stagnalis individuals involved in the study came from our parasite-free laboratory culture, and were reared in the same water and feeding conditions as the experimental groups’. For the experiment, we randomly selected reproductively mature individuals, i.e. individuals with shell length > 20 mm and capable of producing eggs. The shell length (measured with a caliber) corresponds to the distance between the apex and the aperture following the central axis.
We considered biological variability by forming random groups of young adults of Lymnaea stagnalis and Ophryotrocha diadema. It is important to note that rearing conditions were kept uniform in both our L. stagnalis laboratory culture and in the experimental groups involved in the study (i.e. tap water, temperature range 18-22°C, salad as primary source of food). This choice was, indeed, made to minimize potential bias due to individual responses of the subjects to the changing water/feeding conditions. We added these details in 2.3 sub-session of Materials and Methods session.
As for the O. diadema, in our laboratory are present O. diadema strains, cultured for years at the same laboratory conditions: T = 20˚C, artificial sea water salinity = 35 ‰ and they are fed with parboiled spinach. Such conditions were obviously maintained also during the experiment. The only variable factor was the different concentration of BP-3 per group. To this aim, five different treatments were randomly formed of adult virgin hermaphrodites of the same age, no siblings (i. e. offspring (F1) of 24 O. diadema parent pairs (PP)). To control variability eleven replicates per group and control group (individuals set up in the pure artificial sea water) were set up.
We included these information in Materials and Methods session, 2.6. sub-session (Experimental set up for Ophryotrocha diadema).
Results:
15. Reviewer 1: “Make sure all mention of concentrations refers to nominal (not actual) concentrations”.
Authors: at the end of introduction session, we indicated that the tested concentrations of BP-3 are referred to “nominal” concentrations.
16. Reviewer 1: Table 3. This table is confusing when you consider the methodology and approach used. For example, out of the 20 snails you took hemolymph from (and counted 1000 cells) you found a total aggregate of 6 MNi (or 6 cells out of 20000 that you evaluated). If so, then the number of MNi/cell would be 0.0003 MNi/cell (not 0.3MNi/cell as shown). Please check this table carefully.
Authors: we apologize, we forgot to insert ‰ before MNi and NBUDs. In the revised version, Table 3 has been corrected.
17. Reviewer 1: “Mortality data is not presented. This is important, because you must adjust reproductive output to the number of surviving animals in each treatment and/or replicate”.
Authors: We did not observe any deaths for any of the treatments and for Lymnaea stagnalis.
Related to Ophryotrocha, toxic effects of the highest BP-3 concentration (0.2 µg/mL) were observed also in the higher mortality rate (47.27%, 52/110) of individuals: during the experiment we estimated all the individuals present and alive in all the replicates of 5 treatments. At the highest concentration (0.2 µg/mL) the number of individuals decreased from 110 to 52, while in all the rest of the treatments, including controls, the mortality rate was constant (from 110 to 104). We estimated the reproductive output considering only alive individuals and we included mortality data to Results and discussed them in Discussion, as you suggested.
18. Reviewer 1: “Table 4. Please remove this table, as we do not need to see data for each individual snail. Instead please show the Mean body length and SD for each condition at week 0 (start) and Week 4 (end). Carry out statistics to determine if body length was different at the start of the exposure between any of the conditions”.
Authors: as suggested, we modified Table 4 including data about body length before and after treatment and the relative statistic evaluation.
Table 4. Body length (in mm) of L. stagnalis before treatment and after 4 weeks of treatment with different concentrations of BP-3.
|
NC Mean ±SD |
Group 4 0.025 mg/L Mean ±SD |
Group 3 0.05 mg/L Mean ±SD |
Group 2 0.10 mg/L Mean ±SD |
Group 1 0.20 mg/L Mean ±SD |
Before treatment |
32.73±1.55 |
32.45±1.92 |
32.40±1.47 |
32.68±1.78 |
32.60±1.78 |
After 4 weeks of treatment |
49.23±2.83 |
48.60±268 |
48.34±2.44 |
46.39±2.57 ** |
44.23±3.16 * |
Differences |
16.51±2.90 |
16.15±1.98 |
15.94±2.19 |
13.71±2.35 °° |
11.61±2.17 ° |
SD denotes standard deviation; NC negative control; BP-3 benzophenone-3; Whitney test; p <0.001);
*significantly lower compared to NC, 0.025 and 0.05 BP-3 concentrations (ANOVA-test; p <0.001); **significantly lower compared to NC (ANOVA test; p=0.015); °significantly lower compared to NC, 0.025 and 0.05 BP-3 concentrations (Mann-Whitney-test; p <0.001); °° significantly lower compared to NC (Mann-Whitney-test; p=0.005).
19. Reviewer 1: “Table 5 – please state in the legend if there was any mortality. If there is, you must adjust for this in your analysis (eggs/snail)”.
Authors: as discussed at the point 15, for Lymnaea stagnalis the mortality rate was 0 for each treatment. We inserted this information in the legend of Table 5. For Ophryotrocha, we estimated the reproductive output considering only alive individuals. Since in Lymnaea stagnalis the mortality was null for each treatment, we did not modify the legend of the Table 5. However, we modified the legend of the Figure 4, where we added that the reproductive output was estimated only in alive individuals of O. diadema.
20. Reviewer 1: “Like the L. stagnalis experiment, the polychaete exposure is also lacking acclimation and baseline readings, so there is the question of influence of biological variability. The photo of polychaete reproduction (Figure 2) suggests it is quite variable (based on numbers of eggs per cocoon shown)”.
Authors: Related to acclimation, O. diadema has been cultured for more than 30 years at the same laboratory conditions as mentioned above. Moreover, since O. diadema is a simultaneous hermaphrodite, reproductive variability in this species, as well as in L. stagnalis, is influenced by population density (monogamy versus promiscuity). In monogamy egg production is favored, while in promiscuity the egg production decreases because of several factors, mainly the direct competition between individuals (Charnov 1982; Lorenzi et al., 2006).
In the present experiment the population density was constant (and consequently also the reproductive variability), since in all the treatments 10 adult individuals per bowl were set up. In the Figure 2 we wanted just to show two different strains (yellow and white) of O. diadema. However, you promptly noted the different number of eggs in the Figure 2A and 2B. Figure 2A shows two individuals in monogamy, while 2B shows a detail of the mass culture, where the individuals are set up at high population density, therefore in this case, you can see fewer eggs in the cocoon.
Charnov, E.L. 1982. The Theory of Sex Allocation. Princeton University Press, Princeton, NJ, USA
Lorenzi MC, Schleicherová D, Sella G (2006) Life history and sex allocation in the simultaneously hermaphroditic polychaete worm Ophryotrocha diadema: the role of sperm competition. Integr Comp Biol 46:381–389
21. Reviewer 1: “There is a mention of mortality occurring in some treatments in the polychaete exposure (line 303 states 47% mortality). Please include mortality data”.
Authors: We added the mortality at the end of Results.
22. Reviewer 1: “Please explain if the data presented in Figure 4 has adjusted for mortality. If not please do so. I would advise that you replot Figure 4 so that it shows mean cumulative fecundity and SD, as an aggregate (i.e. shows the total number of eggs produced over time for each condition/treatment)”.
Authors: As we mentioned above, we modified the legend of the Figure 4, where we added that the reproductive output was estimated considering only alive individuals. Usually, getting the results at first sight clear for reader, in order to show the mean reproductive output in each group, we report the mean number of eggs produced per treatment (±SD).

Reviewer 2 Report
Santovito et al. analyze the impact of an organic ultraviolet filter, benzophenone-3 (BP3), in a snail and an annelid. They studied the effect in the long-term of different concentrations of BP3 for four weeks. They study the growth effect on the snail and the damage to the eggs in both species. In addition, they study the genotoxicity with micronuclei analysis.
The introduction is informative but missing information about the organisms used (why they were selected? how the life cycle is? how long? etc.). The authors include the information in the material and methods section (points 2.2 and 2.5). However, a couple of sentences in the introduction will help the reader to know the organisms used and the rationalization of the selection.
The material and methods require clarification. First, Lymnea stagnalis cultures, from where do they come? How are they maintained? Why were selected animals with shells bigger than 2 cm? Is it associated with maturity?
Using glass recipients for future exposure would be adequate since the plastic recipients could release additives during the experiment, confusing the results. How many replicates were used? It indicates the number of treatments but does not specify how many replicates were used. In addition, the light-dark regime should be indicated. How was the length measured? It is said that it was measured at the start and end of the experiment but not how it was measured (mantle, shell, from where to where). It is said that tap water was used; are the snails unaffected by the chlorine?
The sample was 20 individuals; are they coming from the same vessel? So, had each 10-L recipient 20 individuals?
Where was the Ophryotrocha diadema obtained? It is not indicated. The culture conditions should be provided, not only exposure conditions. For this species, it is indicated eleven replicates while no replicates were indicated for the snail. How many animals were used in each replicate of O. diadema?
Concerning the exposure, it is unexplained how were prepared the experimental solutions. Were they prepared from a stock? Which was the solvent, if any? Was it used in the control?
In results the table should present the results, from top to down, by increasing concentrations (0-0.025-0.05-0.1-0.2). In the present form, getting the results at first sight is confusing. On the other hand, unless it has some relevance, it makes no sense to present the individual data for the length. It is enough with the mean and standard deviation. Again, the data should be presented to the reader. It is easier to understand them by increasing concentrations from left to right than the opposite.
The discussion fits the results, but an important issue is mentioned here but not in results or material and methods. The authors state in lines 302-303 that there is mortality in the individuals at the highest concentration. It is not previously mentioned. Are they referring to the adults or the eggs? It is unclear. If they refer to adults, it should be mentioned in the results and show the mortality rate. It is relevant in the discussion of the results. Also, it would be a result in case they are referring to eggs, so it means that BP3 affects embryogenesis.
The discussion is missing a paragraph about what means the effects observed for a population of each species. Could it be dangerous and compromise the future? Have been found similar levels of BP3 in natural habitats?
Minor: authors should value use the concentrations in mg/L more than ug/mL. It is common to use them as mg/L, facilitating the reader's comprehension compared to other studies.
Author Response
REVIEWER 2
The authors would like to thank the anonymous reviewer for the careful revision, which helped us improve the manuscript.
In the text, changes have been highlighted in red.
1. Reviewer 2: “The introduction is informative but missing information about the organisms used (why they were selected? how the life cycle is? how long? etc.). The authors include the information in the material and methods section (points 2.2 and 2.5). However, a couple of sentences in the introduction will help the reader to know the organisms used and the rationalization of the selection”.
Authors: In the present study we investigated toxic effects of BP-3, one of the most common UV filters found in sunscreen products. Thus, BP-3, has a grave impact on water quality of aquatic and marine ecosystems and consequently poses risks also to human health. For this purpose, we have chosen to test effects of this compound on two study models present in aquatic environments, one in freshwater (L. stagnalis) and one in the marine environment (O. diadema). Moreover, our study models are both simultaneous hermaphrodites with rather fast reproductive cycles and short lifetime. Thus, both study models are suitable to study genotoxic effects of BP-3 as well as its properties as endocrine disruptor. Detailed information related to the lifespan, lifetime and reproductive strategy of our study models are reported in sections 2.1.2. and 2.1.4.
Yes, we added it in the introduction session, as suggested.
2. Reviewer 2: “The material and methods require clarification. First, Lymnea stagnalis cultures, from where do they come? How are they maintained?”
Authors: L. stagnalis individuals involved in the study came from our parasite-free laboratory culture and were reared in the same water and feeding conditions as the experimental groups’. We included this information in Materials and Methods session, 2.3. sub-session (Experimental set-up for Lymnaea stagnalis)
3. Reviewer 2: “Why were selected animals with shells bigger than 2 cm? Is it associated with maturity?”
Authors: We randomly selected individuals with shell length > 20 mm to avoid any confounding factor involving incomplete sexual maturity. In the revised version of the manuscript, we added these details in Materials and Methods session, 2.3. sub-session (Experimental set-up for Lymnaea stagnalis).
4. Reviewer 2: “Using glass recipients for future exposure would be adequate since the plastic recipients could release additives during the experiment, confusing the results”.
Authors: We acknowledge the concern regarding the use of plastic recipients, and we will consider using glass recipients for future exposures to minimize the risk of release of additives. However, we would like to underline that at room temperature (around 20°C) the release of chemical substances is low and that the same type of container was also used for the control group. Therefore, all groups analyzed in the study had the same “background” of potentially toxic substances released from the container and the same chemical composition of the water. We included these considerations in Materials and Methods session, 2.3. sub-session (Experimental set-up for Lymnaea stagnalis).
5. Reviewer 2: “How many replicates were used? It indicates the number of treatments but does not specify how many replicates were used”.
Authors: A single replicate of 20 individuals per concentration was used. Albeit this could seem a low number, please consider that the Micronucleus Assay (and in particular, the microscopic analyses) requires much time. For this reason, similar studies usually have similar or even lower numbers of individuals per treatment (e.g. Wrisberg et al., 1992; de Vasconcelos Lima et al., 2019).
Wrisberg, M.N.; Bilbo, C.M.; Spliid, H. Induction of micronuclei in hemocytes of Mytilus edulis and statistical analysis. Ecotoxicol. Environ. Saf. 1992, 23, 191-205. https://doi.org/10.1016/0147-6513(92)90058-B
de Vasconcelos Lima, M.; de Siqueira, W.N.; Silva, H.A.M.F.; de Melo Lima Filho, J.; de França, E.J.; de Albuquerque Melo, A.M.M.Cytotoxic and genotoxic effect of oxyfluorfen on hemocytes of Biomphalaria glabrata. Environ. Sci. Pollut. Res. Int. 2019, 26, 3350-3356. https://doi.org/10.1007/s11356-018-3848-3.
6. Reviewer 2: “In addition, the light-dark regime should be indicated. How was the length measured?”
Authors: Although we did not experimentally control or measure the light-dark regime, it was kept uniform for all the treatment groups. We indicated this information in the revision version of the manuscript, in Materials and Methods session, 2.3. sub-session (Experimental set-up for Lymnaea stagnalis).
7. Reviewer 2: “It is said that it was measured at the start and end of the experiment but not how it was measured (mantle, shell, from where to where”.
Authors: The shell length corresponds to the distance between the apex and the aperture following the central axis. We specified this in the revised version, in Materials and Methods session, 2.3. sub-session (Experimental set-up for Lymnaea stagnalis).
8. Reviewer 2: “It is said that tap water was used; are the snails unaffected by the chlorine? “
Authors: in the revised version of the paper, we included data about the chemical analysis of used water. The chemical analysis evidenced the presence of free residual chlorine at concentration of 0.1 mg/L, and chlorides at concentration of 17 mg/L. These elements/compounds could theoretically affect snails. However, we would like to emphasize that L. stagnalis individuals involved in the study came from our parasite-free laboratory culture, and were reared in the same water and feeding conditions as the experimental groups’. During rearing, we didn’t observe abnormal increased mortality or decline in eggs production, or any physiological signs of suffering. Moreover, it is important to note that we used tap water and that rearing conditions were kept uniform in both our L. stagnalis laboratory culture and in the experimental groups involved in the study (i.e. tap water, temperature range 18-22°C, salad as primary source of food). This choice was, indeed, made to minimize potential bias due to individual responses of the subjects to the changing water/feeding conditions. We added these details in 2.3 sub-session of Materials and Methods session.
9. Reviewer 2: “The sample was 20 individuals; are they coming from the same vessel? So, had each 10-L recipient 20 individuals?”
Authors: Tested individuals came from 3 different vessels. Each 10-L recipient contained 20 individuals.
10. Reviewer 2: “Where was the Ophryotrocha diadema obtained? It is not indicated”.
Authors: We utilized individuals of an O. diadema strain derived from individuals collected by Prof. B. Åkesson (University of Stockholm) in 1976 and 1980 in Californian harbours (Long Beach). In order to increase genetic variability and to refresh our laboratory populations, new individuals were added to the strain in 1995 and again in 2001, 2006 and in 2021. These animals came from other laboratory cultures, and were kindly sent by Prof. B. Åkesson, Prof. R. Simonini (University of Modena) and Prof. M. C. Lorenzi (University of Paris). We added this information to Materials and Methods session, 2.5 subsession (Ophryotrocha diadema), as suggested.
11. Reviewer 2: “The culture conditions should be provided, not only exposure conditions”.
Authors: Culture conditions of this species are very simple: T = 20˚C, artificial sea water salinity = 35 ‰, feeding based on parboiled spinach. We added this information to Materials and Methods session, 2.6 subsession (Experimental set up for Ophryotrocha diadema), as suggested.
12. Reviewer 2: “For this species, it is indicated eleven replicates while no replicates were indicated for the snail. How many animals were used in each replicate of diadema?”
Authors: For Lymnaea, we did not make replicates because they are not generally required for genotoxicology studies. This serves to align in vivo studies with in vitro ones, conducted, for example, on human lymphocytes for which no replications are required.
For O. diadema, in each replicate 10 individuals were used. This information is reported in the Table 2.
13. Reviewer 2: “Concerning the exposure, it is unexplained how were prepared the experimental solutions. Were they prepared from a stock? Which was the solvent, if any? Was it used in the control?”
Authors: For Lymnaea stagnalis, in order to obtain the final concentrations of 0.2, 0.1, 0.05 and 0.025 mg/L, we dissolved 1.6, 0.8, 0.4 and 0.2 g of BP-3 in 8 L of water, respectively.
For Ophryotrocha diadema, we prepared a stock solution of BP-3 at the concentration of 2 mg/L. The stock solution was subsequently diluted 10, 20, 40 and 80 times to reach the final concentrations of 0.2, 0.1, 0.05 and 0.025 mg/L used during the experiment.
We inserted this information in materials and method, 2.3 and 2.6 sub-sessions.
The Negative Control was represented by water without BP-3.
14. Reviewer 2: “In results the table should present the results, from top to down, by increasing concentrations (0-0.025-0.05-0.1-0.2). In the present form, getting the results at first sight is confusing”.
Authors: We modified the Table 3 as you suggested.
Table 3. Induction of micronuclei in hemocytes (total cells scored: n=20,000) from L. stagnalis exposed to four different concentrations of BP-3.
BP-3 (mg/L) |
No. of observed MNi |
‰ MNi/total cells scored ± S.D. |
No. of observed NBUDs |
‰ NBUDs/total cells scored ±SD |
BNCs |
‰ BNCs/total cell scored ±SD |
NC |
6 |
0.300±0.470 |
20 |
1.000±1.026 |
8 |
0.400±0.60 |
0.025 |
8 |
0.400±0.598 |
21 |
1.050±0.945 |
10 |
0.500±0.83 |
0.050 |
9 |
0.450±0.686 |
23 |
1.150±1.040 |
13 |
0.650±0.99 |
0.100 |
18 |
0.900±0.852 *** |
28 |
1.400±1.536 |
12 |
0.600±1.19 |
0.200 |
25 |
1.250±0.910 * |
44 |
2.200±1.963 ** |
10 |
0.500±0.61 |
15. Reviewer 2: “On the other hand, unless it has some relevance, it makes no sense to present the individual data for the length. It is enough with the mean and standard deviation. Again, the data should be presented to the reader. It is easier to understand them by increasing concentrations from left to right than the opposite”.
Authors: We modified Tables 4 and 5 as you suggested
16. Reviewer 2: “The discussion fits the results, but an important issue is mentioned here but not in results or material and methods. The authors state in lines 302-303 that there is mortality in the individuals at the highest concentration. It is not previously mentioned. Are they referring to the adults or the eggs? It is unclear. If they refer to adults, it should be mentioned in the results and show the mortality rate. It is relevant in the discussion of the results. Also, it would be a result in case they are referring to eggs, so it means that BP3 affects embryogenesis”.
Authors: we refer to both, adults and eggs. Eggs: After Week 2 of treatment, all eggs exposed to the two highest BP-3 concentrations (0.2 and 0.1 μg/mL) were found to be degenerated: eggs were laid, but they did non develop and consequently got dissolved. Adults: Toxic effects of the highest BP-3 concentration (0.2 μg/mL) were observed in the higher mortality rate (47.27%, 52/110) of individuals: at the end of the experiment, we estimated the number of individuals present and alive in each replicate of the 5 treatments. At the highest concentration (0.2 μg/mL) the number of individuals decreased from 110 to 52, while in all the rest of the treatments, including controls, the mortality rate was constant (from 110 to 104). The estimation of the mortality is mentioned in Methods 2.1.5. However, we added this information also to Results.
Related to the embryogenesis, we added to Discussion that the highest concentration of BP-3 could affect embryogenesis as you suggested.
17. Reviewer 2: “The discussion is missing a paragraph about what means the effects observed for a population of each species. Could it be dangerous and compromise the future?”
Authors: It is difficult to predict which impacts could these concentrations of BP-3 generate on the natural populations of the two organisms, or on other similar species, also considering the possible synergistic effect with different xenobiotics present in natural environments, including other endocrine disruptors and pesticides. However, we would like to emphasize that, in the present work, we analyzed the effects of BP-3 at genomic level and on two fitness parameters, body lenght and egg production, that could have consequences at population level. Indeed, if a xenobiotic induces genomic instability and a decrease in the body grow rate and number of eggs laid, then the number of adults able to reproduce will decrease as a consequence. Numerous studies demonstrated that growth inhibition in aquatic invertebrates may be associated with reduction in reproductive outputs. For example, Bessa da Silva et al. (2016) showed the negative effect of a dietary exposure to herbicide on the reproductive output of Daphnia magna. Similarly, Henry et al. (2022) demonstrated the negative effects of fluoxetine on reproduction of freshwater snail Physa acuta, whereas Cole et al. (2015) evidenced the negative impact of micropalstics on function and fecundity in the marine copepod Calanus helgolandicus, with consequences on the population dynamics of these species.
We added these considerations in Discussion
18. Reviewer 2: “Have been found similar levels of BP3 in natural habitats?”
Authors: In the introduction session, we included the following statement: “BP-3 has been detected at the maximum concentrations of 125 ng/L in freshwater [10, 11], whereas for seawater concentrations values of 216 ng/L for Italy [12] and 3317 ng/L for Spain [13] were recorded.”
19. Reviewer 2: “Minor: authors should value use the concentrations in mg/L more than ug/mL. It is common to use them as mg/L, facilitating the reader's comprehension compared to other studies”.
Authors: in the revised version of the paper, we substituted ug/mL with mg/L, as suggested.

Round 2
Reviewer 1 Report
The revised manuscript is much improved and clearer following revision. In the future, I would urge the authors to consider undertaking analytical chemistry of water, and to avoid the use of plastics where it is practical to do so. Replication (and a baseline period) is always good practice, when measuring endpoints like reproduction. This will provide greater confidence that changes observed are caused by the chemical.
The English in some of the new text shown in red could be improved slightly.
Author Response
Reviewer 1
- The revised manuscript is much improved and clearer following revision. In the future, I would urge the authors to consider undertaking analytical chemistry of water, and to avoid the use of plastics where it is practical to do so. Replication (and a baseline period) is always good practice, when measuring endpoints like reproduction. This will provide greater confidence that changes observed are caused by the chemical.
Authors: We thank the reviewer for the positive comments and suggestions. We agree that, for the evaluation of life-history endpoints like reproduction or body growth, subdividing the samples in replicates is a good practice. We will certainly consider it for our future research with model animal organisms.
Thank you again.
- Comments on the Quality of English Language. The English in some of the new text shown in red could be improved slightly.
Authors: we revised English, as suggested.

Reviewer 2 Report
Concerning the responses, most of the comments were satisfactorily answered. However, a couple of commentaries are needed. First, many laboratories use tap water but dechlorinate it to favor the organisms. There are different ways to do it; even leaving the water with aeration will remove chlorine.
The other comment is about the replicates with L. stagnalis. The authors have used 20 animals in one exposure. Why did they not divide the animals into three or four sets to do three replicates? They would have the same 20 animals and the technical replicates needed for the experiment.
Author Response
Reviewer 2
Concerning the responses, most of the comments were satisfactorily answered. However, a couple of commentaries are needed.
- First, many laboratories use tap water but dechlorinate it to favor the organisms. There are different ways to do it; even leaving the water with aeration will remove chlorine.
Authors: Dear author, you are right. However, we would like to specify that the animals were raised for 1 month under the same conditions and therefore all with the same water. Furthermore, the breeding containers were open to allow the animals to breathe. Therefore, probably, during this period at least some of the chlorine evaporated.
- The other comment is about the replicates with stagnalis. The authors have used 20 animals in one exposure. Why did they not divide the animals into three or four sets to do three replicates? They would have the same 20 animals and the technical replicates needed for the experiment.
Authors: For Lymnaea, we decided not to divide the animals into 3 or 4 replicates because this procedure is not generally required for genotoxicology studies. Moreover, we also wanted to align in vivo studies with in vitro ones conducted on human lymphocytes, for which no replications are required. Instead, for the Ophryotrocha, as there is no genotoxic data, we decided to do the replications, as required.
In the revised version of the paper, in materials an methods session, in 2.3 subsession (Experimental set-up for Lymnaea stagnalis), we included this statement.
However, we agree with reviewer that for life-history endpoints, like reproduction or body growth, subdividing the samples in replicates could be a good practice that we will certainly adopt for our future works with model animal organisms.
